# The Effects of Prenatal Exposure to Pregabalin on the Development of Ventral Midbrain Dopaminergic Neurons

**DOI:** 10.3390/cells11050852

**Published:** 2022-03-01

**Authors:** Walaa F. Alsanie, Majid Alhomrani, Ahmed Gaber, Hamza Habeeballah, Heba A. Alkhatabi, Raed I. Felimban, Sherin Abdelrahman, Charlotte A. E. Hauser, Adeel G. Chaudhary, Abdulhakeem S. Alamri, Bassem M. Raafat, Abdulwahab Alamri, Sirajudheen Anwar, Khaled A. Alswat, Yusuf S. Althobaiti, Yousif A. Asiri

**Affiliations:** 1Department of Clinical Laboratories Sciences, The Faculty of Applied Medical Sciences, Taif University, P.O. Box 11099, Taif 21944, Saudi Arabia; m.alhomrani@tu.edu.sa (M.A.); a.alamri@tu.edu.sa (A.S.A.); 2Centre of Biomedical Sciences Research (CBSR), Deanship of Scientific Research, Taif University, P.O. Box 11099, Taif 21944, Saudi Arabia; a.gaber@tu.edu.sa; 3Department of Biology, College of Science, Taif University, P.O. Box 11099, Taif 21944, Saudi Arabia; 4Department of Medical Laboratory Technology, Faculty of Applied Medical Sciences in Rabigh, King Abdulaziz University, Jeddah 21589, Saudi Arabia; hhabeeballah@kau.edu.sa; 5Department of Medical Laboratory Sciences, Faculty of Applied Medical Sciences, King Abdulaziz University, Jeddah 21589, Saudi Arabia; halkhattabi@kau.edu.sa (H.A.A.); faraed@kau.edu.sa (R.I.F.); chaudhary@kau.edu.sa (A.G.C.); 6Center of Excellence in Genomic Medicine Research (CEGMR), King Abdulaziz University, Jeddah 21589, Saudi Arabia; 7King Fahd Medical Research Centre, Hematology Research Unit, King Abdulaziz University, Jeddah 21589, Saudi Arabia; 8Center of Innovation in Personalized Medicine (CIPM), 3D Bioprinting Unit, King Abdulaziz University, Jeddah 21589, Saudi Arabia; 9Laboratory for Nanomedicine, Division of Biological and Environmental Science and Engineering (BESE), King Abdullah University of Science and Technology (KAUST), Thuwal 23955, Saudi Arabia; sherin.abdelrahman@kaust.edu.sa (S.A.); charlotte.hauser@kaust.edu.sa (C.A.E.H.); 10Computational Bioscience Research Center (CBRC), King Abdullah University of Science and Technology, Thuwal 23955, Saudi Arabia; 11Department of Radiological Sciences, College of Applied Medical Sciences, Taif University, P.O. Box 11099, Taif 21944, Saudi Arabia; bassemraafat@tu.edu.sa; 12Department of Pharmacology and Toxicology, College of Pharmacy, University of Hail, Hail 55211, Saudi Arabia; a.alamry@uoh.edu.sa (A.A.); si.anwar@uoh.edu.sa (S.A.); 13Department of Internal Medicine, School of Medicine, Taif University, P.O. Box 11099, Taif 21944, Saudi Arabia; k.alswat@tu.edu.sa; 14Department of Pharmacology and Toxicology, College of Pharmacy, Taif University, P.O. Box 11099, Taif 21944, Saudi Arabia; ys.althobaiti@tu.edu.sa; 15Addiction and Neuroscience Research Unit, Taif University, P.O. Box 11099, Taif 21944, Saudi Arabia; 16Department of Clinical Pharmacy, College of Pharmacy, Taif University, P.O. Box 11099, Taif 21944, Saudi Arabia; yasiri@tu.edu.sa

**Keywords:** pregabalin, neuropathic pain, embryonic neurons, ventral midbrain dopaminergic neurons

## Abstract

Pregabalin is widely used as a treatment for multiple neurological disorders; however, it has been reported to have the potential for misuse. Due to a lack of safety studies in pregnancy, pregabalin is considered the last treatment option for various neurological diseases, such as neuropathic pain. Therefore, pregabalin abuse in pregnant women, even at therapeutic doses, may impair fetal development. We used primary mouse embryonic neurons to investigate whether exposure to pregabalin can impair the morphogenesis and differentiation of ventral midbrain neurons. This study focused on ventral midbrain dopaminergic neurons, as they are responsible for cognition, movement, and behavior. The results showed that pregabalin exposure during early brain development induced upregulation of the dopaminergic progenitor genes *Lmx1a* and *Nurr1* and the mature dopaminergic gene *Pitx3*. Interestingly, pregabalin had different effects on the morphogenesis of non-dopaminergic ventral midbrain neurons. Importantly, our findings illustrated that a therapeutic dose of pregabalin (10 μM) did not affect the viability of neurons. However, it caused a decrease in ATP release in ventral midbrain neurons. We demonstrated that exposure to pregabalin during early brain development could interfere with the neurogenesis and morphogenesis of ventral midbrain dopaminergic neurons. These findings are crucial for clinical consideration of the use of pregabalin during pregnancy.

## 1. Introduction

Pregabalin (Lyrica) was the first drug approved by the Food and Drug Administration for the treatment of postherpetic neuralgia and diabetic neuropathy [1]. There is much evidence, both in pre-clinical and clinical settings, claiming that pregabalin has demonstrated effectiveness in the management of neuropathic pain (NP) [2]. The NP-relieving and other neurological effects of pregabalin, in combination or as a monotherapy, have been observed in a dose-dependent manner in clinical settings [3,4]. The use of pregabalin has major advantages, such as easy use, high tolerance, and relative reliability in patients suffering from NP [5]. Pregabalin is now considered the successor of gabapentin and has demonstrated good efficacy in various models of incisional injury, inflammatory injury, and NP [2]. The neuroprotection offered by pregabalin may be why pregabalin offers pain-attenuating effects since NP is due to neuronal damage. Several animal pain models demonstrated analgesic effects of pregabalin in distinct pain models, such as NP [2]. The analgesic action of pregabalin is exerted through its antagonistic activity against voltage-dependent calcium channels, where it binds to its α-2-δ subunit [6]. Prescriptions for pregabalin have increased significantly over the past few years [6]. In the United States alone, prescriptions rose from 39 million in 2012 to 64 million in 2016, increasing annual costs from USD 2–4.4 billion [7]. In the UK, the number increased by 350% over the five years from 2008 to 2013 [8]. In 2017, the cost incurred for the 6.2 million pregabalin prescriptions in England was USD 440 million [6].

Pregabalin has been misused, which might cause unavoidable effects. Studies regarding safety in pregnant women have been an issue, as pregabalin is considered the last treatment option for NP. It is still unclear whether pregabalin at therapeutic doses causes abnormalities in the fetus and hampers its development [9]. Previously, we showed that pregabalin induces behavioral sensitization through the dopamine reward system, which suggests that pregabalin could have an effect on dopaminergic neurons [10].

Ventral midbrain dopaminergic (vmDA) neurons are an important part of the neuronal population that plays a major role in motor activity and cognition. Because of the lack of concrete evidence regarding the impact of pregabalin on the fetal brain, especially vmDA neurons, we used embryonic ventral midbrain (VM) neurons to investigate whether pregabalin could alter their developmental cues.

## 2. Materials and Methods

### 2.1. Isolation of Primary Mouse Embryonic Ventral Midbrain Dopaminergic Neurons

This study followed the international guidelines for the use of animals in research and the experiments were approved by the King Abdulaziz University (KAAU) and Taif University Ethics Committees (1-441-133). The experimental protocol is illustrated in Figure 1.

Embryos were isolated from time-mated Swiss mice (Animal House Center, King Fahad Medical Research Center, KAAU, Jeddah, Saudi Arabia). Animals were time-mated overnight, and the visualization of a vaginal plug on the following morning was considered embryonic day (E) 0.5. The VMs of E12.5 mouse embryos were dissected in ice-cold L15 media (Thermofisher, Waltham, MA, USA). The telencephalon–mesencephalon boundary and isthmic organizer were cut to isolate the midbrain and cortical tissues. The ventral third of the midbrain tissue was dissected and used to enrich the dopaminergic population in the culture. The isolated VMs were incubated in 0.05% trypsin (Thermofisher, Waltham, MA, USA) and 0.1% DNase (Stem Cell Technologies, Cambridge, MA, USA) diluted in Ca/Mg-free Hank’s Balanced Salt Solution (HBSS) (Thermofisher, Waltham, MA, USA) for 15 min at 37 °C. The tissues were washed (three times in HBSS media) and resuspended in N2 media (consisting of a 1:1 mixture of F12 medium and Minimum Essential Medium supplemented with 1 mM glutamine, 1 mg/mL bovine serum albumin, 15 mM HEPES, 6 mg/mL glucose, 1% penicillin/streptomycin, and 1% N2 supplement) (all N2 media components from Thermofisher, Waltham, MA, USA). Primary neurons were cultured for three days in vitro, depending on the experiments (demonstrated in the sections below) prior to assessments.

### 2.2. Three-Dimensional Neuronal CELL Culture and Pregabalin Treatment

Three-dimensional in vitro cultures are advantageous because they mimic the normal development in vivo. The 3D cultures established in our study were used for the viability, ATP release, morphogenesis, and quantitative PCR experiments. The establishment of 3D cultures was as follows. Freshly isolated primary embryonic mouse E12.5 vmDA neurons were seeded at 6 × 10^4^ cells/well in 96-well cell culture plates for 3D cultures. To establish 3D cultures, we used the short peptide IIZK, as previously described [11]. IIZK was prepared in 1 × DPBS. The weighed IIZK peptide was first resuspended in a volume of nuclease-free sterile water that was half of the required final volume. As the gelation was almost instantaneous once DPBS was added, DPBS was only added inside the cell culture well. A suitable volume of the previously resuspended peptide in water was added to the culture well, and an equivalent volume of 2 × DPBS was added to enhance the gelation process. A peptide base was first added to each well to prevent cells from coming into contact with the plastic surface. The plates were then incubated for 5 min to ensure complete gelation. A 3D construct was then constructed on top of the peptide base. Peptides were added and the desired number of cells in 2 × DPBS was added to an equivalent volume and mixed briefly with the peptide. The plates were incubated again for 2–3 min before adding cell culture media. N2 medium was then carefully added to the culture plates. The cells were incubated at 37 °C and 5% CO_2_ for 72 h. Pregabalin (Sigma, St. Louis, MO, USA) was prepared according to the manufacturer’s instructions in sterile 1 × PBS. Then, 10 μM pregabalin was added to the pregabalin-treated group directly after seeding the cells, while an equal volume of sterile 1 × PBS was added to the control groups. 

### 2.3. vmDA Neuronal Viability and ATP Release Assessment

It is crucial to investigate the effect of any drug on the viability and metabolic activity of the target cells. We assessed the viability of VM neurons and ATP release in response to pregabalin treatment after 3 days of culture. To assess the viability of the VM neurons in the control and pregabalin-treated cultures, we used alamarBlue™ Cell Viability Reagent (Thermo Fisher, USA). We performed the experiment according to the manufacturer’s instructions. After preparing the well plates, the fluorescence was measured using a PHERAstar FS plate reader (BMG LabTech, Ortenberg, Germany).

ATP release was assessed using the CellTiter-Glo^®^ 3D Cell Viability Assay (Promega, Madison, Wisconsin, WI, USA) as an indicator of the metabolic activity status of the cells. Briefly, CellTiter-Glo^®^ Reagent was added in a volume equal to that of the cell culture medium and mixed thoroughly by pipetting up and down 10 times to break down the 3D construct comprising the cells and hydrogel. Plates were then incubated at room temperature for 25 min, and the luminescent signal was read using a PHERAstar FS plate reader (BMG LabTech, Germany). 

### 2.4. Immunocytochemistry 

Mouse embryonic vmDA neurons were fixed after three days in culture using 4% paraformaldehyde (Santa Cruz Biotechnology, Paso Robles, CA, USA) and stored at 4 °C in 1 × PBS until staining was performed. Primary antibodies against mouse neuron-specific class III beta-tubulin (TUJ1) (Promega (Madison, WI, USA), G7121) and tyrosine hydroxylase (TH) (Abcam, Cambridge, UK), ab112) were used. Fixed cultures were incubated with primary antibodies that were diluted as follows: TUJ1 (1:1500) and TH (1:500) in blocking buffer (5% goat serum, 0.3% Triton-X, and 0.2% sodium azide) overnight at room temperature. After removing the primary antibodies, the cells were incubated for 1 h at room temperature in a blocking buffer. This was followed by the addition of anti-mouse Alexa 488 and goat anti-rabbit IgG H&L (Alexa Fluor^®^ 555) (Abcam, ab150078). Secondary antibodies were diluted in blocking buffer (1:200) and incubated for 2 h at room temperature. Subsequently, the cells were incubated for 5 min in DAPI (Thermo Fisher Scientific, D1306) diluted in 1 × PBS, and the wells were washed and kept in 1 × PBS. Imaging was performed using a DMi8 inverted fluorescent microscope (Leica, Wetzlar, Germany). 

### 2.5. Morphogenetic Analysis

The exposure of neurons to certain drugs during development could alter their morphogenesis, which could affect their connectivity with their targets in the brain. The effects of pregabalin on the morphogenesis of VM neurons were assessed in stained cultures. The number of neurites, total length of neurites, length of dominant neurites, and number of branches were analyzed as previously described [12], where LAS X software was used for the analysis (Leica, Germany). Overlapping neurites and those shorter than 20 µm were excluded to avoid bias. Data obtained from the pregabalin-treated cultures were normalized to those of the control group. Subsequently, the data were expressed as a percentage change from the control, which was considered to be 100%.

### 2.6. Quantitative PCR 

The differentiation of neurons is the main developmental process to generate functional neurons. It is fundamental to assess the effect of any drugs on the differentiation of the target cells. One of the assessment tools is used to investigate the expression of the key genes involved in the development of this population. Therefore, RNA was isolated after 3 days in culture using the RNeasy Plus Universal Mini Kit (Qiagen, Hilden, Germany), Cat No. 73404) following the manufacturer’s instructions. To ensure efficient homogenization of the cells, TissueLyser II (Qiagen, Hilden, Germany) was used as recommended in the RNeasy kit protocol. RNA was isolated from VM neurons in pregabalin-treated and control cultures. As a negative control, RNA was isolated from mouse tissues other than those of the brain. The primer sequences of the selected genes were listed in Table 1.

The RT-PCR StepOne System and Data Assist software were used to generate raw cycle threshold (CT) data for the negative control (negative tissue) and experimental groups for both housekeeping/reference (*GAPDH*) genes and target genes (*Th*, *Nurr1*, *Lmx1a*, *En1*, *Pitx3*, *Drd2*, *Dat*, and *Bdnf*) in triplicate. We normalized the CT of the target gene to the CT of the reference gene and ∆CT of the test sample to the ∆CT of the control sample before analysis using the ∆∆CT method. The ΔCT (CT target gene − CT reference gene) value was calculated for each sample, ΔΔCT (ΔCT test sample − ΔCT control sample) value was calculated for comparative groups, and finally, we calculated the relative quantification (Rq = 2^−∆∆CT^) and fold change (log2FC) to evaluate the expression of target genes under different experimental conditions. The Rq values of each gene were compared across all samples of the five groups and *p*-values were calculated for the identification of significantly expressed genes. 

### 2.7. Statistical Analysis

Student’s *t*-test was performed using GraphPad Prism v 8.1.2 (San Diego, California, CA, USA), and all quantitative data are expressed as mean ± SEM with the significance set at *p* < 0.05. 

## 3. Results

### 3.1. Pregabalin Disrupted Metabolic Activity of vmDA Neurons 

Previous studies showed that pregabalin has several neurotoxic effects with long-term use [13,14]. These negative effects have a significant deleterious impact on neurodevelopment and brain function in neurological and cognitive disorders [15]. The viability of neurons was not affected as compared to that of the controls (Figure 2A), which confirmed that this dose was not toxic to the cells. We observed that pregabalin disrupted the metabolic activity of the vmDA neurons by significantly decreasing the ATP generation compared to that in the control culture (Figure 2B). These results demonstrated that pregabalin interfered with the electron transport chain, which resulted in decreased ATP generation. 

### 3.2. Pregabalin Affected the Morphogenesis of vmDA Neurons

The effects of pregabalin on the morphogenesis of vmDA neurons were assessed in stained cultures. We observed morphological changes in neurite length (Figure 3A) and dominant neurite length (Figure 3B); however, we could not see any major differences in the number of branches (Figure 3C) or the number of neurites (Figure 3D) in the pregabalin-treated cultures vs. the control cultures (Figure 3E,F). These results suggested that exposure to pregabalin affected the morphogenesis and differentiation potential of neurons, which was further investigated by examining the gene expression involved in the differentiation cues.

### 3.3. Pregabalin Affected the Morphogenesis of vm Non-DA Neurons

To understand whether the effects of pregabalin mentioned earlier meant a general effect on all VM neurons or it was exclusive to vmDA neurons, the effects of pregabalin on the morphogenesis of vm non-DA neurons (TH−/TUJ1+) were assessed in the stained cultures. We observed morphological changes in the number of branches (Figure 4C) and the number of neurites (Figure 4D); however, we did not observe significant differences in the total neurite length (Figure 4A) or the dominant neurite length (Figure 4B), between the pregabalin-treated cultures and control cultures (Figure 4E,F). These findings confirmed that pregabalin exposure affected the differentiation and morphogenesis of dopaminergic and non-dopaminergic neurons in the VM.

### 3.4. Pregabalin Induced the Upregulation of Key Dopaminergic-Related Genes in vmDA Neurons

Many key regulators of neurogenesis were uncovered, such as early DA fate-determinant genes, such as *Lmx1a/b*, and maturation genes, such as *Wnt5a*, *Pitx3*, and *Th.* However, these genes fail to encompass all the processes involved in vmDA development [16,17,18]. *Lmx1a/b* is important for vmDA neuronal generation [19,20]. We observed a significant change in the expression of *Lmx1a* in the pregabalin-treated cultures (Figure 5A). The high expression of *Lmx1a* suggested that this may be a mechanism for pregabalin increasing the differentiation potential/morphogenesis of vmDA neurons. These neuronal lineage-specific transcription factors regulate the expression of several downstream genes and determine the morphological, functional, and physiological identities of mDA neurons [19]. It is also important to emphasize the significantly high expression of *Nurr1* in pregabalin-treated cultures (Figure 5B). *Lmx1a*, *Nurr1*, and *Mash1* are part of a minimal transcription factor mixture, generating vmDA neurons directly from human and murine fibroblasts without reverting to a progenitor cell stage [21]. *Lmx1a* and *Lmx1b* have been closely associated with neurological disorders, and it is important to know their exact role in maintaining vmDA neurons [22]. Several studies have revealed that transcription factors, such as *Nurr1*, *En1*, *Pitx3*, and *Lmx1a*, play an active role in the early development of vmDA neurons and are important for maintaining adult phenotypic neuronal identity [23]. In addition, previous studies reported that *Lmx1a* activates *Th,* which helps differentiate vmDA neurons [24]. In the pregabalin-treated cultures, we observed a significant difference in the expression of *Pitx3* (Figure 5D). However, genes such as *Th*, *Chl1*, and *En1* were not significantly altered (Figure 5C,E,F). The expression of *Pitx3* revealed that pregabalin influenced the early maturation potential of vmDA neurons. We found that pregabalin treatment significantly altered the expression of *Lmx1a*, *Nurr1*, and *Pitx3*, while no change was observed in the *En1*, *Th*, and *ChI1* genes. These findings suggested that pregabalin signaled to the *Lmx1a*/*Nurr1*/*Pitx3* pathway for neuronal differentiation. It was shown that *Nurr1* has downstream target genes that are important for the maturation and functionality of vmDA neurons, such as *Bdnf*, *Dat*, and *Drd2* [25,26,27,28]. We demonstrated that the expression of these three target genes was altered by pregabalin treatment. There was a significant upregulation in *Bdnf*, *Dat*, and *Drd2* in pregabalin-treated cultures in comparison with the control culture (Figure 5G,H,I). Recent studies revealed that several genes are important in vmDA development, such as *Chl1* (close homolog to L1) [24]; however, its expression in our pregabalin-treated cultures was not significantly different from that of the control cultures (Figure 5F).

## 4. Discussion

During development, the crosstalk between differentiation and proliferation dictates the size of each nucleus within the central nervous system (CNS). The fate of vmDA neurons is generally determined by the multiple interplay activities of intrinsic and extrinsic factors. Neurons respond to many signals for their neuronal connectivity within the current cell cycle, establishing a final resting place and adopting a differentiated state. Many of these processes in the late and early developmental stages of VM neurons have been detected. This neuronal population is important for behavior, cognition, and motor activity. Our previous study suggested that the dopaminergic system could be affected by pregabalin administration [10]. In the present study, we identified that pregabalin at therapeutic doses (10 μM) had no impact on the survivability of neurons [29]. However, it caused a decrease in ATP release in VM neurons. Our present study revealed that pregabalin exposure at an early stage interferes with the neurogenesis and morphogenesis of vmDA neurons. These studies are important regarding making decisions on the clinical use of pregabalin during pregnancy. Transcription factors, such as *Lmx1a*, *Pitx3*, and *Nurr1*, are essential for midbrain regional identity, and we observed significantly high expression with exposure to pregabalin, which suggested that pregabalin might not be safe for pregnant women.

Pregabalin is a gabapentin-like alkylated derivative of aminobutyric acid (GABA)-approved medications for neuropathic syndromes, fibromyalgia, partial-onset seizures, and generalized anxiety disorder in several parts of the world. Off-label use has been investigated for restless leg syndrome and other psychiatric illnesses, such as cyclic mood disorders. Pregabalin has also been used recreationally [30]. A study assessed patients who received pregabalin, where they observed that most of them used it recreationally (63%), and among them, 60% were off-label use [31]. Due to a lack of safety studies in pregnancy, pregabalin is considered the last treatment option for various neurological diseases, such as NP, for pregnant women [32]. Recently, the safety of pregabalin use during pregnancy was only assessed in a few studies, and this has not been confirmed. Studies also suggested that pregabalin easily permeates the BBB and placental barrier in mice, rats, and monkeys, and this is of obvious importance for drugs that influence the CNS during development [33]. Therefore, further studies are needed to evaluate the safety aspects of pregabalin in pregnant women and for fetus health. In the same context, we conducted this study to investigate the effects of prenatal exposure to pregabalin on the development of vmDA neurons, which play an essential role in drug addiction, cognition, regulation of emotions and rewards, and voluntary movements [34]. We exposed primary mouse embryonic vmDA neurons to pregabalin to assess its toxic effects on the viability of vmDA. We observed that pregabalin treatment did not alter vmDA neuron viability, suggesting that pregabalin was not toxic at a dose of 10 μM. This observation was consistent with those of earlier studies in which the potential toxic effect of pregabalin on cellular models was assessed [35]. A previous study investigated individual antiepileptic drugs related to pregnancy. The study concluded that antiepileptics relative to lamotrigine, valproic acid, and carbamazepine were associated with a smaller head circumference [36]. Another research study demonstrated that pregabalin administration during pregnancy is associated with adverse gestational outcomes. The study concluded that there is no clarity in using pregabalin during pregnancy. They proposed that it can be selected based on risk to benefit ratio [37]

Meanwhile, we observed that pregabalin treatment at a dose of 10 μM disrupted vmDA neuronal metabolic activity, mediated by a decrease in ATP release. These results suggested that pregabalin interferes with the electron transport chain, resulting in decreased ATP generation. Next, we assessed the impact of pregabalin on the morphogenesis of vmDA neurons and found that the length of the total and dominant neurites significantly increased after pregabalin treatment; however, the numbers of neurites and branches were not significantly changed. Meanwhile, the opposite effects were observed in the case of non-vmDA neuron culture, where pregabalin significantly altered the number of neurites and their branches but not the length of the neurites. 

Several recent transcriptional studies have revealed the involvement of various genes in the development of vmDA. These genes include *Lmx1a*, *Lmx1b*, *En1*, *Nurr1*, *Th*, and *Pitx3* [24,38,39]. Studies have reported that *Lmx1a* has functional significance in postnatal life and continues to be expressed in postmitotic precursors and differentiating neurons [22]. *Lmx1a* activates *Nurr1* [22], subsequently activating the *Th* that helps differentiate vmDA neurons [24]. In contrast, *Lmx1a* activates *Rspo2*, which is involved in the regulation of *Pitx3* [40]. Considering these observations, we further explored the effects of pregabalin on these genes to determine whether it can affect the expression and perhaps thereby alter the differentiation of vmDA neurons. We found that pregabalin treatment significantly increased the expression of *Lmx1a*, *Nurr1*, and *Pitx3*, while no change was observed in the *En1*, *Th*, and *ChI1* genes. These findings suggest that pregabalin follows the *Lmx1a*/*Nurr1*/*Pitx3* pathway for neuronal differentiation. It was demonstrated that Nurr1 controlled the expression of several downstream targets, such as *Bdnf*, *Drd2*, and *Dat* [25,26,27,28]. *Bdnf* is a trophic factor for vmDA neurons, which is important for their survival [41]. On the other hand, *Drd2* and *Dat* are crucial for the development and functionality of mature vmDA, respectively [28,42,43]. In the present study, we showed that *Bdnf*, *Drd2*, and *Dat* are upregulated significantly in the presence of pregabalin in culture, which could have been due to the upregulation of *Nurr1*. It was shown that *Pitx3*, which is also upregulated in the presence of pregabalin, directly activates *Dat* in coordination with *Nurr1* [42]. Interestingly, the upregulation of *Drd2* could explain the increase in TH+ neurites length observed in the pregabalin-treated cultures. It was shown that stimulation of Drd2 receptors induced axons elongation in dopaminergic neurons [42]. The disruption in the expression of these genes is associated with different neurological and psychological problems [44,45,46]. 

## 5. Conclusions

Our data suggested that pregabalin abuse in pregnant women, or even at therapeutic doses, may impair fetal development. We used primary mouse embryonic VM neurons to investigate whether prenatal exposure to pregabalin can impair fetal brain development. This study focused on vmDA neurons, which are responsible for cognition, movement, and behavior. The current study demonstrated that exposure to pregabalin during early brain development could interfere with the neurogenesis and morphogenesis of vmDA neurons. Additionally, several genes were identified that allowed pregabalin have an impact on vmDA. These findings are crucial for the clinical use of pregabalin during pregnancy. Further, an in vivo study should be conducted to further explore the effects of pregabalin on vmDA, which may contribute to understanding the pregabalin-mediated development of vmDA neurons. 

## Figures and Tables

**Figure 1 cells-11-00852-f001:**
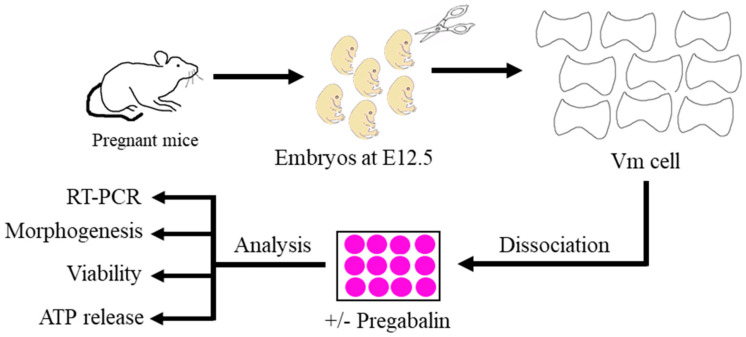
Illustration showing the experimental design for the present study.

**Figure 2 cells-11-00852-f002:**
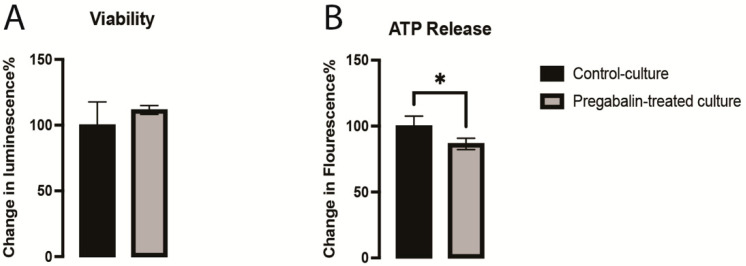
Demonstration of viability (**A**) and ATP release (**B**) in the pregabalin-treated and control cultures. Data are represented as mean ± SEM, n = 7 experiments. * *p* < 0.05.

**Figure 3 cells-11-00852-f003:**
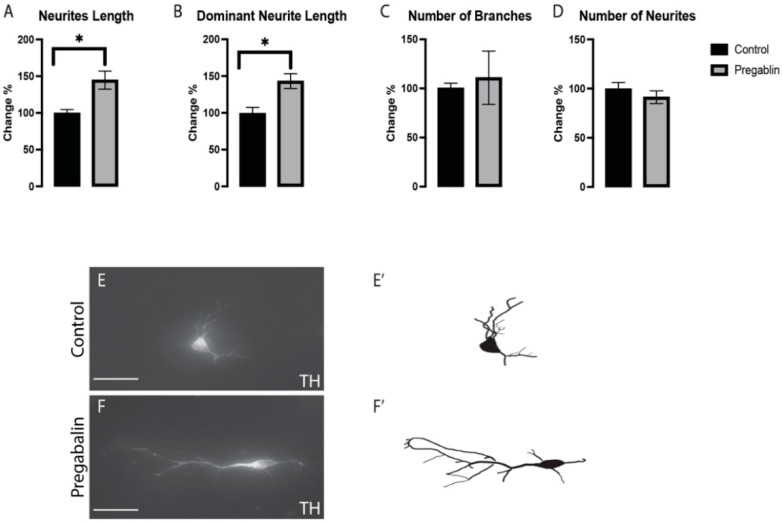
Pregabalin induced the elongation in TH+ vmDA neurons’ total neurite length (**A**) and dominant neurite length (**B**), but there was no significant effect on the number of branches (**C**) or neurites (**D**). Representative images and illustrations for vmDA neurons immunolabeled with TH in both groups; control (**E**,**E’**) and pregabalin-treated (**F**,**F’**) groups showed an increase in neurite elongation in response to pregabalin exposure. Scale bar: 50 μm. Data are represented as mean ± SEM, n = 4 experiments. * *p* < 0.05.

**Figure 4 cells-11-00852-f004:**
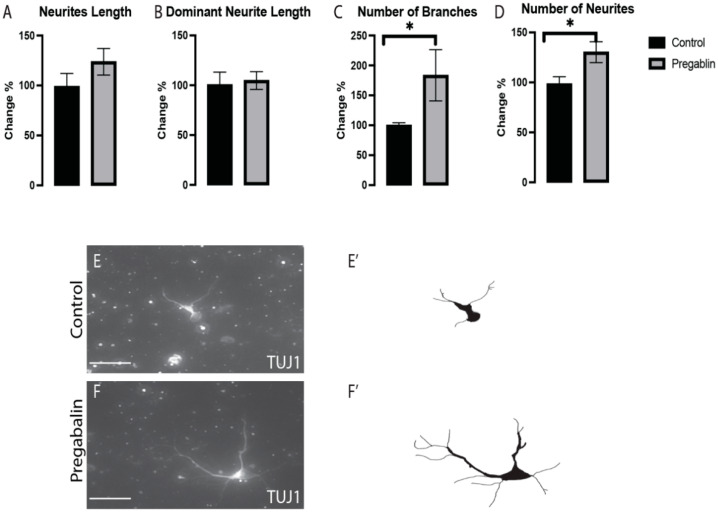
Pregabalin exposure did not affect the total neurite length (**A**) nor the dominant neurite length (**B**), while the number of branches (**C**) and neurites (**D**) were significantly increased in VM non-dopaminergic neurons (TUJ1+/TH−). Representative images and illustrations for VM neurons immunolabeled with TUJ1 in both groups; control (**E**,**E’**) and pregabalin-treated (**F**,**F’**) groups showed increases in the numbers of branches and neurites in response to pregabalin exposure. Scale bar: 50 μm. Data are represented as mean ± SEM, n = 4 experiments. * *p* < 0.05.

**Figure 5 cells-11-00852-f005:**
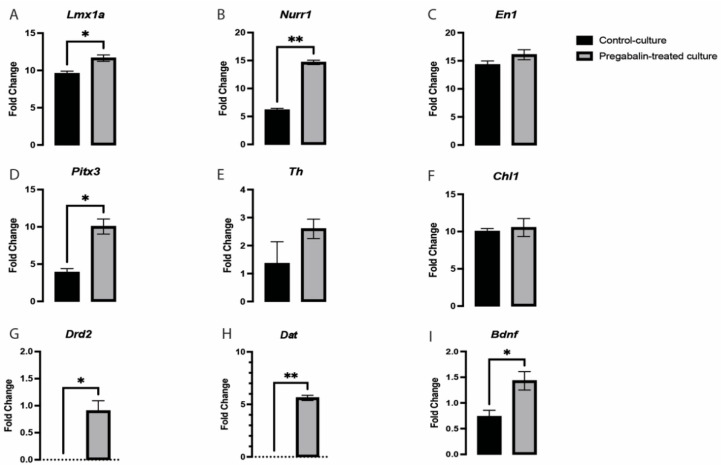
Pregabalin exposure caused a significant upregulation in the expression of *Lmx1a* (**A**), *Nurr1* (**B**), *Pitx3* (**D**), *Drd2* (**G**), *Dat* (**H**), and *Bdnf* (**I**), but there was no significant change in the expression of *En1* (**C**), *Th* (**E**), or *Chl1* (**F**). Data are represented as mean ± SEM, n = 3 experiments. * *p* < 0.05, ** *p* < 0.01.

**Table 1 cells-11-00852-t001:** Gene-specific primer pair sequences that were used in the RT-PCR.

Gene Name	Primer Sequence (5′ to 3′)
*GAPDH*	F-primer: TGAAGGTCGGAGTCAACGGA R-primer: CCAATTGATGACAAGCTTCCCG
*Th*	F-primer: TGAAGGAACGGACTGGCTTC R-primer: GAGTGCATAGGTGAGGAGGC
*Nurr1*	F-primer: GACCAGGACCTGCTTTTTGA R-primer: ACCCCATTGCAAAAGATGAG
*Lmx1a*	F-primer: GAGACCACCTGCTTCTACCG R-primer: GCACGCATGACAAACTCATT
*En1*	F-primer: TCACAGCAACCCCTAGTGTG R-primer: CGCTTGTCTTCCTTCTCGTT
*Pitx3*	F-primer: CATGGAGTTTGGGCTGCTTG R-primer: CCTTCTCCGAGTCACTGTGC
*Chl1*	F-primer: TGGAATTGCCATTATGTGGA R-primer: CACCTGCACGTATGACTGCT
*Dat*	F-primer: TTGCAGCTGGCACATCTATC R-primer: ATGCTGACCACGACCACATA
*Drd2*	F-primer: CTCAACAACACAGACCAGAAT R-primer: GAACGAGACGATGGAGGA
*Bdnf*	F-primer: ACTATGGTTATTTCATACTTCGGTT R-primer: CCATTCACGCTCTCCAGA

## Data Availability

All the data supportive stated results are existing in the manuscript.

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
