# Peer review of "The Effects of Prenatal Exposure to Pregabalin on the Development of Ventral Midbrain Dopaminergic Neurons"

_cells, 2022, doi:10.3390/cells11050852_

Round 1

Reviewer 1 Report

Overall, this is an interesting research article in which Alsanie et al. evaluate the effects of the exposure to pregabalin during early brain development in order to test the safety of this treatment during pregnancy. They found that pregabalin interfere in the neurogenesis and morphogenesis of ventral midbrain dopaminergic neurons, finding that are crucial for the clinical use of pregabalin during pregnancy.  

Minor comments:

  1. Please describe in the figure legends, panel E and F of figures 3 and 4.
  2. Abbreviations in text and figure legends. In general, you should review that you use the word or phrase in full when you refer the term for the first time followed by the abbreviation in parentheses and thereafter use the abbreviation only. For instance, if you abbreviate ventral midbrain in line 79, you don’t have to abbreviate again in line 92. The same for dopaminergic ventral midbrain (vmDA) in line 75 and 106 and after you should put always the abbreviature. Check also neuropathic pain, central nervous system…
  3. Please add references in line 188 or remove this sentence from the text. You should include:

Fatma M. Elgazzar, Walaa Sayed Elseady, Amal SAF Hafez. Neurotoxic effects of pregabalin dependence on the brain frontal cortex in adult male albino rats, NeuroToxicology, Volume 83,2021, Pages 146-155,ISSN 0161-813X,https://doi.org/10.1016/j.neuro.2021.01.004.

Taha, S.H.N., Zaghloul, H.S., Ali, A.A.E.R. et al. The neurotoxic effect of long-term use of high-dose Pregabalin and the role of alpha tocopherol in amelioration: implication of MAPK signaling with oxidative stress and apoptosis. Naunyn-Schmiedeberg's Arch Pharmacol 393, 1635–1648 (2020). https://doi.org/10.1007/s00210-020-01875-5

  1. Others references that could be interesting to include to improve introduction or discussion are:

Relation of in-utero exposure to antiepileptic drugs to pregnancy duration and size at birth.Andrea V. Margulis, Sonia Hernandez-Diaz,Thomas McElrath,Kenneth J. Rothman,Estel Plana,Catarina Almqvist,Brian M. D’Onofrio,Anna Sara Oberg. Published: August 5, 2019.https://doi.org/10.1371/journal.pone.0214180

Andrade C. Safety of Pregabalin in Pregnancy. J Clin Psychiatry. 2018 Oct 2;79(5):18f12568. doi: 10.4088/JCP.18f12568. PMID: 30289631.

Author Response

We would like to thank the reviewer for his/her valuable comments. Please find below the actions that have been taken in regard the suggestions.

  1. Please describe in the figure legends, panel E and F of figures 3 and 4.

Thanks for your comment. We have added the missing description for the panels in figure 3 and 4 along with the scale bar.

  1. Abbreviations in text and figure legends. In general, you should review that you use the word or phrase in full when you refer the term for the first time followed by the abbreviation in parentheses and thereafter use the abbreviation only. For instance, if you abbreviate ventral midbrain in line 79, you don’t have to abbreviate again in line 92. The same for dopaminergic ventral midbrain (vmDA) in line 75 and 106 and after you should put always the abbreviature. Check also neuropathic pain, central nervous system…

We thank the reviewer for these comments. We have fixed the problem with the abbreviations for ventral midbrain in line 92. For vmDA, we have fixed the problem in lines 106 and 310. We have looked also for neuropathic pain and CNS and fixed the problem in lines 302 (for the former), 280 and 306 (for the lateral).

  1. Please add references in line 188 or remove this sentence from the text. You should include:

Fatma M. Elgazzar, Walaa Sayed Elseady, Amal SAF Hafez. Neurotoxic effects of pregabalin dependence on the brain frontal cortex in adult male albino rats, NeuroToxicology, Volume 83,2021, Pages 146-155,ISSN 0161-813X,https://doi.org/10.1016/j.neuro.2021.01.004.

Taha, S.H.N., Zaghloul, H.S., Ali, A.A.E.R. et al. The neurotoxic effect of long-term use of high-dose Pregabalin and the role of alpha tocopherol in amelioration: implication of MAPK signaling with oxidative stress and apoptosis. Naunyn-Schmiedeberg's Arch Pharmacol 393, 1635–1648 (2020). https://doi.org/10.1007/s00210-020-01875-5

We thank the reviewer for these comments. . We have fixed the problem and inserted the references suggested.

  1. Others references that could be interesting to include to improve introduction or discussion are:

Relation of in-utero exposure to antiepileptic drugs to pregnancy duration and size at birth.Andrea V. Margulis, Sonia Hernandez-Diaz,Thomas McElrath,Kenneth J. Rothman,Estel Plana,Catarina Almqvist,Brian M. D’Onofrio,Anna Sara Oberg. Published: August 5, 2019.https://doi.org/10.1371/journal.pone.0214180

Andrade C. Safety of Pregabalin in Pregnancy. J Clin Psychiatry. 2018 Oct 2;79(5):18f12568. doi: 10.4088/JCP.18f12568. PMID: 30289631.

We thank the reviewer for these comments.  We have fixed the problem and improved our discussion.

We have added in the discussion section “A previous study investigated individual antiepileptic drugs related to pregnancy. The study concluded that antiepileptics relative to lamotrigine, valproic acid, and carbamazepine were associated with smaller head circumference [31]. Another research study demonstrated that pregabalin administration during pregnancy is associated with adverse gestational outcomes. The study concluded that there is no clarity in using pregabalin during pregnancy. They proposed that it can be selected based on risk to benefit ratio[32]” in the lines 339-345

Reviewer 2 Report

In the manuscript cell-1591738 by Alsanie et al. the authors analyzed the effect of pregabalin on E12.5 midbrain primary cultures.

Pregabalin is a calcium channel blocker and could reduce presynaptic levels of dopamine in the brain and cause an increase of Parkinsonian symptoms. In the introduction and discussion sections this aspect is not explained and is not clear why dopaminergic neurons are analyzed.

The manuscript could be accepted for publication after major revision and by rectifying some of the typographical mistakes and citation errors stated as below:

  • Line 36-37 “Therefore, pregabalin abuse in pregnant women, even at therapeutic doses, may impair fetal development. We used primary mouse embryonic neurons to investigate whether prenatal exposure to pregabalin can impair fetal brain development”. To investigate whether prenatal exposure to pregabalin can impair fetal brain development the female should be treated, and later primary embryonic cultures should be made.  Using only E12.5 mouse primary culture you are investigating the effect of pregabalin only on the differentiation and the genes involved in the midbrain dopaminergic identity, such as Nurr1, Pitx3 and TH.
  • To elucidate the results section, the rationale of the experiment should be explained.
  • When the 3D neuronal cell cultures have been used? For ATP release assessment?
  • 3-days in vitro is a short time, which is the effect at long-time? Is it toxic?
  • How many TH+ cells do you have in culture?
  • Could be interesting also analyze the Nurr1 target genes (BDNF, DAT, DRD2, VMAT2).
  • Pregabalin influence the maturation efficiency and the branch complexity of neurons, is the effect of Pregabalin tested on mature cultures, for instance E15-E16?
  • Florescent images should be presented.

Minor points:

Line 120 Co2

Line 188 Add refs

Author Response

We would like to thank the reviewer for his/her valuable comments. Please find below the actions that have been taken in regard the suggestions.

We have justified the use of vmDA neurons in our present study in lines 76-79 in the introduction as the following “Previously, we have shown that pregabalin induces behavioural sensitization through the dopamine reward system, which suggests that pregabalin could have an effect on dopaminergic neurons.”

And we have also added the following in the discussion section (Lines 309-311) “This neuronal population is important for behaviour, cognition and motor activity. Our previous study suggested that the dopaminergic system could be affected by pregabalin administration .”

  • Line 36-37 “Therefore, pregabalin abuse in pregnant women, even at therapeutic doses, may impair fetal development. We used primary mouse embryonic neurons to investigate whether prenatal exposure to pregabalin can impair fetal brain development”. To investigate whether prenatal exposure to pregabalin can impair fetal brain development the female should be treated, and later primary embryonic cultures should be made.  Using only E12.5 mouse primary culture you are investigating the effect of pregabalin only on the differentiation and the genes involved in the midbrain dopaminergic identity, such as Nurr1, Pitx3 and TH.

We thank the reviewer for his comment. We have modified the sentence (line 37-39) to be as following:

“We used primary mouse embryonic neurons to investigate whether exposure to pregabalin can impair the morphogenesis and differentiation of ventral midbrain neurons”

  • To elucidate the results section, the rationale of the experiment should be explained.

We thank the reviewer for the comment. We have added the following in response to the comment:

Line137-138 “It is crucial to investigate the effect of any drug on the viability and metabolic activity on the target cells.”

Lines 167-168 “The exposure of neurons to certain drugs, during development, could alter their morphogenesis which could affect their connectivity with their targets in the brain”

Lines 178-181 “The differentiation of neurons is the main developmental process to generate functional neurons. It is fundamental to assess the effect of any drugs on the differentiation of the target cells. One of the assessment tool is investigating the expression of the key genes involved in the development of this population.”

  • When the 3D neuronal cell cultures have been used? For ATP release assessment?

We thank the reviewer for the valuable comment. We have the added the following in the lines 113-116 “3D in vitro cultures are advantageous because they are mimicking the normal development in vivo. The 3D cultures established in our study have been used for the viability, ATP release, morphogenesis and quantitative PCR experiments. The establishment of 3D culture was as the following.”

  • 3-days in vitro is a short time, which is the effect at long-time? Is it toxic?

Thanks very much for the valuable comment. We normally test the effect of drug/protein in mouse embryonic neuronal cultures for three-four days. This is the time frame we rely on and from our previous investigations we believe it is sufficient to decide if there is an effect. In addition, the mouse neuronal cultures, generally, are difficult to remain viable in long time cultures, opposite to human neuronal cultures. We have listed below examples of few studies used similar experimental design to the one we have used here.

Alsanie WF, Penna V, Schachner M, Thompson LH, Parish CL. Homophilic binding of the neural cell adhesion molecule CHL1 regulates development of ventral midbrain dopaminergic pathways. Sci Rep. 2017 Aug 24;7(1):9368. doi: 10.1038/s41598-017-09599-y. PMID: 28839197; PMCID: PMC5570898.

Blakely BD, Bye CR, Fernando CV, et al. Wnt5a regulates midbrain dopaminergic axon growth and guidance. PLoS One. 2011;6(3):e18373. Published 2011 Mar 31. doi:10.1371/journal.pone.0018373

Bye CR, Rytova V, Alsanie WF, Parish CL, Thompson LH. Axonal Growth of Midbrain Dopamine Neurons is Modulated by the Cell Adhesion Molecule ALCAM Through Trans-Heterophilic Interactions with L1cam, Chl1, and Semaphorins. J Neurosci. 2019;39(34):6656-6667. doi:10.1523/JNEUROSCI.0278-19.2019

Chathurini V. Fernando, Julianna Kele, Christopher R. Bye, Jonathan C. Niclis, Walaa Alsanie, Brette D. Blakely, Jan Stenman, Brad J. Turner, and Clare L. Parish.Stem Cells and Development.Sep 2014.1991-2003.

  • How many TH+cells do you have in culture?

We thank the reviewer for the question. We know from our previous studies that TH+ neurons in VM cultures, isolated from E12.5 and E11.5, are around 10% and 5%, respectively, of the total neurons. Listed below a couple of our previous studies for further info.

Chathurini V. Fernando, Julianna Kele, Christopher R. Bye, Jonathan C. Niclis, Walaa Alsanie, Brette D. Blakely, Jan Stenman, Brad J. Turner, and Clare L. Parish.Stem Cells and Development.Sep 2014.1991-2003.

Alsanie WF, Penna V, Schachner M, Thompson LH, Parish CL. Homophilic binding of the neural cell adhesion molecule CHL1 regulates development of ventral midbrain dopaminergic pathways. Sci Rep. 2017 Aug 24;7(1):9368. doi: 10.1038/s41598-017-09599-y. PMID: 28839197; PMCID: PMC5570898.

  • Could be interesting also analyze the Nurr1 target genes (BDNF, DAT, DRD2, VMAT2).

We thanks very much the reviewer for the suggestions. We have performed additional RT-PCR for the suggested genes. We have added the results in figure 5 for all the genes, except Vmat2 gene as the expression was very low in both control and pregabalin treated cultures, so we couldn’t find any difference. We have also added few sentences in the results sections in the lines 300-305 as the following

“It was shown that Nurr1 has downstream target genes that are important for the maturation and functionality of vmDA neurons, such as Bdnf, Dat and Drd2 [25–28] We demonstrate that the expression of these three target genes is altered by pregabalin treatment. There was a significant upregulation in Bdnf, Dat and Drd2 in pregabalin-treated cultures in comparison with control culture (Figure 5G, H, I).”

Also we have added few sentences in the discussion section in the lines 381-406 as the following

It was demonstrated that Nurr1 control the expression of several downstream targets, such as Bdnf, Drd2 and Dat [25–28]. Bdnf is a trophic factor for vmDA neurons, which is important for their survival [41] On the other hand, Drd2 and Dat are crucial for the development and functionality of mature vmDA, respectively [28, 42, 43] In the present study, we show that Bdnf, Drd2 and Dat, are upregulated significantly, in the presence of pregabalin in culture, which could be due to the upregulation of Nurr1. It has been shown that Pitx3, which is also upregulated in the presence of pregabalin, directly activates Dat in coordination with Nurr1 [42]. Interestingly, the upregulation of Drd2 could explain the increase in TH+ neurites length observed in the pregabalin-treated cultures. It was shown that stimulation of Drd2 receptors induced axons elongation in dopaminergic neurons [42]. The disruption in the expression of these genes is associated with different neurological and psychological problems[44–46]

  • Pregabalin influence the maturation efficiency and the branch complexity of neurons, is the effect of Pregabalin tested on mature cultures, for instance E15-E16?

Thanks for the comment. We know from our previous study that E12.5 is the ideal timepoint for detecting the effects on morphogenesis and maturation for vmDA neurons. At later time points, vmDA become mature and do not respond to external cues for detecting morphogenetic changes. Thus, this was the reason why we have not investigated the effect of pregabalin on later timepoints for the morphogenesis.

  • Florescent images should be presented.

Thanks very much for the comment. All the images presented in our study are florescent images, but we are showing them in black and white color mode as they are single staining images. We believe that black and white are clearer for observing the neurites growing from the neurons. Schematic images are also added to clarify the morphology of the representative neurons shown in our figures.

Minor points:

Line 120 Co2

Line 188 Add refs

Thanks for your comments. We have fixed the errors as per the reviewer instructions.

Reviewer 3 Report

Alsanie et al. Investigate the effect of pregabalin on midbrain dopaminergic neurons. The topic is interesting but some questions should be addressed before publication.

  • I suggest adding some data about the electrophysiological features (for example: firing rate, rhythmicity of firing etc..) of dopaminergic neurons treated with pregabalin vs control condition
  • What can the authors deduce from the increase of Lmx1a, Nurr1 and Pitx3 expression? How could it affect the development of dopaminergic neurons? what effect could it have on the development of the nervous system of the embryos?
  • Do the authors detect changes in the number of non-DA neurons?
  • At line 188 there is a refuse, please remove it
  • Section 3.4 is too long, please move some parts in the discussion section
  • Authors should provide the primer sequence in the material and method section or as supplementary

Author Response

We thank very much the reviewer for the valuable comments. Please find our response as below:

  • I suggest adding some data about the electrophysiological features (for example: firing rate, rhythmicity of firing etc..) of dopaminergic neurons treated with pregabalin vs control condition

Thanks very much for the valuable comments. We totally agree with the reviewer regarding the importance of investigating the functionality of treated cultures. However, the techniques for these experiments are not available at the moments unfortunately. 

  • What can the authors deduce from the increase of Lmx1a, Nurr1 and Pitx3 expression? How could it affect the development of dopaminergic neurons? what effect could it have on the development of the nervous system of the embryos?

We thank the reviewer for the valuable comment. We have added more RT-PCR experiments and investigated the targets genes of Nurr1 and Pitx3. We have updated figure 5 and added the following from lines 281 to 407:

“It was demonstrated that Nurr1 control the expression of several downstream targets, such as Bdnf, Drd2 and Dat [25–28]. Bdnf is a trophic factor for vmDA neurons, which is important for their survival [41] On the other hand, Drd2 and Dat are crucial for the development and functionality of mature vmDA, respectively [28, 42, 43] In the present study, we show that Bdnf, Drd2 and Dat, are upregulated significantly, in the presence of pregabalin in culture, which could be due to the upregulation of Nurr1. It has been shown that Pitx3, which is also upregulated in the presence of pregabalin, directly activates Dat in coordination with Nurr1 [42]. Interestingly, the upregulation of Drd2 could explain the increase in TH+ neurites length observed in the pregabalin-treated cultures. It was shown that stimulation of Drd2 receptors induced axons elongation in dopaminergic neurons [42]. The disruption in the expression of these genes is associated with different neurological and psychological problems[44–46]”

  • Do the authors detect changes in the number of non-DA neurons?

Thanks for the comments. We have not detect any changes in the numbers of Tuj1+/Th- neurons in this study.

  • At line 188 there is a refuse, please remove it

Thanks very much for the comment. We have removed it and added references.

  • Section 3.4 is too long, please move some parts in the discussion section

We thank the reviewer for the valuable comment. We have followed the instructions and shortened 3.4 Section and moved some parts to the discussion.

  • Authors should provide the primer sequence in the material and method section or as supplementary.

We thank the reviewe for the valuble comment. We listed all gene primers that used in RTPCR in Table 1.

Round 2

Reviewer 2 Report

The manuscript includes the changes required and can be accepted in the present form.

Reviewer 3 Report

The authors addressd the reviewer's comments